# Low back pain in the Bangladeshi adult population: a cross-sectional national survey

Muhammad Shoaib Momen Majumder ,[1] Ferdous Hakim ,[2]
Iftekhar Hussain Bandhan ,[1] Mohammad Abdur Razzaque,[3]
Ahmad Zahid-Al-Quadir,[4] Shamim Ahmed,[1] Minhaj Rahim Choudhury ,[1]
Syed Atiqul Haq,[1] MM Zaman [2]

[1]Rheumatology, Bangabandhu Sheikh Mujib Medical University, Dhaka, Bangladesh
[2]World Health Organization Bangladesh, Dhaka, Bangladesh
[3]Medicine, Chittagong Medical College, Chittagong, Bangladesh
[4]Medicine, Sylhet MAG Osmani Medical College, Sylhet, Bangladesh

**Correspondence to**
Dr Muhammad Shoaib Momen Majumder;
momen.shoaib@gmail.com

## ABSTRACT

**Objective** Low back pain (LBP) is a common musculoskeletal disorder. This study aims to determine the residence-specific and sex-specific prevalence and the factors associated with LBP in Bangladesh.

**Methods** The study subjects (aged ≥18 years) were identified from 20 primary sampling units of the national census following a cross-sectional multistage stratified sampling design. We considered the mechanical type of LBP for this study. A Bangla version of the modified Community Oriented Programme for Control of Rheumatic Disorders questionnaire was used. A team of trained field workers, rheumatology residents and rheumatologists collected the data. Analysis was done using weighted data.

**Results** Two thousand subjects were approached, but 1843 could be screened. Among them, 561 had musculoskeletal disorders, and 343 were diagnosed with LBP. The weighted prevalence of LBP was 18.5% (95% CI: 11.8% to 25.2%) and age-standardised prevalence of LBP was 19.4% (95% CI: 14.0% to 24.8%), which was higher in women (27.2%, 19.3% to 35.1%) than men (14.0%, 8.7% to 19.3%). The prevalence persistently increased from age group 18–34 years (10.5%, 5.7 to 15.4) to ≥55 years (27.8%, 16.1% to 39.5%). People with no education had the highest prevalence (31.3%, 22.3% to 40.4%). The prevalence did not differ between urban and rural residential locations. Four factors were significantly associated with LBP: age (adjusted odds ratio: 2.4, 95% CI: 1.7 to 3.4), female sex (2.2, 1.5 to 3.3), absence of formal education (2.3, 1.6 to 3.3) and hypertension (1.7, 1.1 to 2.6).

**Conclusion** LBP is a common problem in Bangladeshi adults. The factors identified are age, female sex, no formal education and hypertension. These should be addressed adequately to prevent and treat LBP.

## STRENGTHS AND LIMITATIONS OF THIS STUDY

⇒ We report the weighted prevalence of low back pain by sociodemographic characteristics, comorbidities, disability and work loss, and identified factors associated with patients with back pain, for the first time in Bangladesh.
⇒ All the diagnoses were made by rheumatology residents and expert rheumatologists in the field.
⇒ Some diagnoses of evolving rheumatological conditions might lack validity because of lack of quality laboratory facilities in the field.
⇒ The sample size calculation is based on combined prevalence of musculoskeletal disorders that warrant cautious interpretation of the results because of inadequate sample size, especially when split into reporting domains.
⇒ Recall period for determining work loss was 12 months which might induce bias.

Around 11%–12% of the population become disabled due to LBP.[1] It causes substantial personal, social and financial burdens globally.[1] In the USA, LBP is the second most frequent cause for a physician consultation.[5] LBP is ranked globally as the topmost cause of disability as it affects mostly working-age people.[6] It accounted for 60.1 million disability-adjusted life-years in 2015.[7] There was a significant increase of LBP by 54% since 1990, and the highest escalation took place in the low-income and middle-income countries (LMICs).[7] Disability from LBP is a primary concern for the LMICs, specially in Bangladesh where manual labour—rickshaw pulling, day labourers, house maids, work exposure to lifting of heavy weight during their day-to-day activities and so on—is common.[7] The scope for job switching is restricted in resource-constraint countries.

LBP has multisectorial health outcomes like a lower quality of life, poorer self-reported health, depression and more

## INTRODUCTION

Low back pain (LBP) is one of the most frequent medical problems globally.[1] It is defined as pain, stiffness or muscle tension localised below the costal margin and above the inferior gluteal folds.[2] Up to 84% of adults suffer from LBP at some point in life.[3] The prevalence of chronic LBP is about 23%.[4]

workspace absenteeism.[8] As a result, LBP has become an important cause of sick leave and early retirement among the working population.[9] In the USA, approximately 149 million workdays are lost due to LBP, leading to an estimated loss of 100–200 billion US dollars per year.[10] Non-specific LBP is the the most common of all causes of LBP.[4] Non-specific LBP is defined as LBP not particularly attributable to specific aetiology like malignancy, infection, fracture, inflammatory condition, radiculopathy or cauda equina syndrome.[4]

Although high in most studies, there is a difference in LBP prevalence in various epidemiological studies. The estimated lifetime prevalence was 84.1% in a Canadian study,[11] 70% in Denmark[12] and 59% in the UK.[13] In Iran, the prevalence of LBP was 29.3%.[14] The estimated prevalence of LBP in India ranged between 42% and 83%.[15 16] A recent cross-sectional, community-based, epidemiological study conducted in Northern India yielded an estimated lifetime prevalence of 47% in man 57% in women.[17] A Community Oriented Programme for Control of Rheumatic Disorders (COPCORD) survey in Bangladesh published in 2005 showed 6.6%, 9.9% and 9.2% prevalence of LBP in the rural, urban slum and affluent urban areas, respectively.[18] A cross-sectional national study in Bangladesh in 2015 showed LBP was the top-ranking musculoskeletal disorder (MSD) with a prevalence of 18.6%.[19] We have further analysed the data from the 2015 study and report the population weighted prevalence according to sociodemographic factors, comorbid conditions, disability and work loss due to LBP, and identify the factors associated with LBP.

## METHODS

A detailed description of the methodology is beyond the scope of this article and is described elsewhere.[19] Adults aged 18 years or more comprised the study population through a household level multistage stratified cross-sectional survey. The sampling frame was based on the 2001 Bangladesh Census.[20] Based on a point prevalence of MSD and with a design effect of 1.5 and 85% response rate for four reporting domains (man–woman, urban–rural), the calculated sample size was 1978, which was rounded to 2000. It was stratified into seven divisions of rural (Mauza) and urban (Mahalla) areas. Twenty (8 urban and 12 rural) primary sampling units (PSUs) were selected. The first 100 households were consecutively included from each PSU, where even numbers were assigned as man and odd numbers as woman households. In each household, the single respondent was identified from a list of eligible household members with the help of a Kish table. Data were collected in November and December of 2015 (figure 1).

A detailed manual was prepared before the training for this survey and was used by all field staff. All investigators and the WHO technical team coordinated and conducted the training. The modified COPCORD questionnaire was the survey tool.[21] The English version of the first part of the questionnaire was translated to Bangla, then adapted according to the guideline of Beaton et al,[22] validated by Chassany's Method[23] and administered by the interviewers. Data were collected for 6 days from each PSU. There were two recall visits to ensure participation. The research physician interviewed the suspected respondents for MSDs. A subject was considered a positive respondent if he/she reported pain in muscles, bones, joints or any part of the body (musculoskeletal system) during the preceding 7 days. Subjects who were taking pain medications like non-steroidal anti-inflammatory drugs or steroids were considered as positive respondent even if they did not report pain on those 7 days. All positive respondents were interviewed and examined by the research physicians. Internationally accepted criteria were used for the diagnosis of the diseases. For the conditions without any internationally accepted criteria, relevant investigations and clinical judgement of the research physician was used. The final diagnoses were checked and verified by a rheumatologist during their visit to the respective PSUs.

LBP group of disorders were operationally defined as mechanical type back pain that included non-specific LBP and lumbar spondylosis. Considering the limitation of differentiating investigation in the field, we did not classify LBP beyond this. LBP duration was classified into three groups—acute: up to 6 weeks, subacute: 6–12 weeks and chronic that persists beyond 12 weeks.[24] Respondents with pain in the muscles, bones, joints or any part of the body (musculoskeletal symptom) during the preceding 7 days or on pain medication with no pain were considered as positive respondents. The research physicians interviewed and thoroughly examined all 'positive' respondents. Data on physical activity were calculated into metabolic equivalent tasks-minutes per week using the STEPwise Surveillance of noncommunicable disease risk factors (STEPS) protocol and divided into quartiles.[25] The fourth quartile was labelled as strenuous physical activity.

The study participants were divided into three subgroups as per age in years: 18–34, 35–54 and 55–99. We considered ownership of household asset items (electricity, television, refrigerator, etc) for constructing wealth index. In addition, the type of main material used for the roof of the main house (cement, tin and katcha such as bamboo/thatched/straw/gunny, etc) was also included in the model. A principal component analysis was used to create standardised factor scores for each of the items. The total scores for the respondents were calculated and categorised those into quartiles for description from one (lowest household wealth) to four (highest household wealth).[19]

A validated Bangla version of the Bangla Health Assessment Questionnaire-Disability Index (HAQ-DI) was used for the disability score. For determining work loss, the recall period was 12 months.[26] Random capillary blood glucose was measured by using glucometer (Accu-Check Germany). Using height (metres) and weight (kilograms)

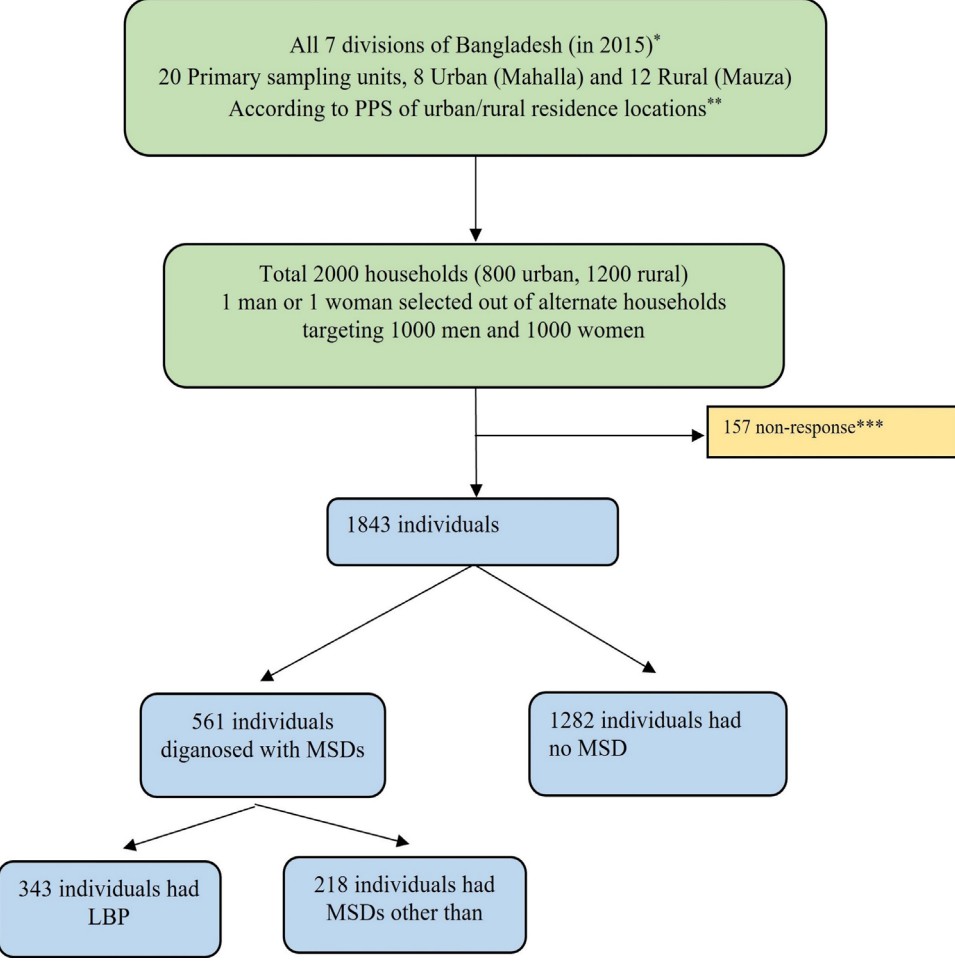

**Figure 1** Flow chart on the selection of patients with LBP from the national survey on musculoskeletal conditions in Bangladesh (2015). *Eight divisions from September 2015. Randomly 15 districts were selected out of 64. **PPS indicates population proportion to size. ***Out-migration, broken house, locked house, no tenant, out of home, refusal. Two recall visits were done if the selected house was locked and the person chosen was not available at home at the time of the interviewer's visit. In case of non-participation after the second recall visit, the targeted household/individual was declared non-respondents. Adapted from Ahmad Zahid-Al-Quadir et al.[21] LBP, low back pain; MSDs, musculoskeletal disorders.

measurements, we calculated body mass index (BMI) (weight (kg)/height (metre)$^2$). Waist circumference was measured by horizontally placing a measuring tape above the iliac crest. Diabetes was defined as random blood glucose ≥11.1 mmol/L or the use of antidiabetic medications. Obesity was defined as a body mass index of ≥25 kg/m$^2$.[27]

**Statistical analysis**

The survey data were entered in and cleaned using Microsoft Excel. We have weighted[28] the data to reflect population frame of Bangladesh for the year 2015. Base weight—for the sampled population—was calculated using probability of selection of respondents among the eligible number of members of household in a cluster defined by divisions (7), age groups (3) and sexes (2). The base weight was adjusted with non-response weights separately for men and women by three age groups. The final weight was generated after calibration to frame population (2015) in domains by division, sex and age groups. Analysis was done using the final weights.

Statistical analysis was done using Epi Info V.7.1.5.2 and in SPSS V.20.0. Continuous variables were categorised before analysis as appropriate. We estimated the prevalence of LBP with 95% CIs. The prevalence was segregated by residence (urban/ rural) and sex (man/ woman). Non-parametric test (Kruskal-Wallis H test) was used to analyse data that were not normally distributed. Whenever we encountered an unweighted respondent size of <25, the confidence intervals were suppressed. Age-standardisation of prevalence estimates was made for global comparison using the WHO World Population 2000–2025.[29] Factors were checked for association with LBP by comparing LBP with no MSD through 2×2 tables. Univariate logistic regression analysis was done to obtain unadjusted ORs. All statistically significant relationships (p<0.05) were entered into a model for logistic regression analysis. The adjusted ORs and their 95% confidence limits were calculated to identify the strength of association of LBP factors. A detailed description of categorisation and analysis of other variables was described elsewhere.[19]

**Table 1** Characteristics of the respondents with LBP by unweighted and weighted numbers, musculoskeletal disease survey 2015

| Sociodemographic characteristics | Unweighted sample (n=1843) Number (per cent) | Weighted* sample (N=94 794 164) Number (per cent) |
|---|---|---|
| | 1843 (100.0) | 94 794 164 (100.0) |
| Sex | | |
| Men | 892 (48.4) | 41 553 976 (43.8) |
| Women | 951 (51.6) | 53 240 188 (56.2) |
| Residence | | |
| Urban | 716 (38.9) | 27 772 657 (29.3) |
| Rural | 1127 (61.2) | 67 021 507 (70.7) |
| Age, years | | |
| 18–34 | 711 (38.6) | 41 343 470 (43.6) |
| 35–54 | 774 (42.0) | 35 278 850 (37.2) |
| 55–99 | 358 (19.4) | 18 171 844 (19.2) |

*Weighted to projected population of Bangladesh from 2001 Population Census Frame of Bangladesh Bureau of Statistics.
LBP, low back pain.

## Ethics approval and consent to participate
Ethical guidelines, as outlined by the Declaration of Helsinki, were followed throughout the study.[30] Ethical clearance was obtained from the Institutional Review Board of Bangabandhu Sheikh Mujib Medical University (ID 1100). Informed written consent was obtained from the respondents in Bangla as per Institutional Review Board's guidelines.

## Patient and public involvement
Patient or the public were not involved in the design, or conduct, or reporting, or dissemination plans of this study.

## RESULTS
## Characteristics of respondents
In this nationally representative study, 2000 adults 18 years or older were approached, and 1843 (92.2%) agreed to participate.[19] The mean age of the participants was 40.5 (SD 14.7) years, and 51.6% were women. A total of 561 (30.4%) had some type of MSDs. LBP was the most common among MSDs (18.6%, unadjusted), followed by knee osteoarthritis (7.3%) and soft-tissue rheumatism (5.2%). Among the inflammatory rheumatic diseases, the common conditions were rheumatoid arthritis (1.6%) and spondyloarthritis (1.3%).

Table 1 shows the changes brough in by the weighting procedure on the unweighted sample. The weighted percentages are more in line with the projected Population Frame[20] from which the study sample was drawn.

## Characteristics of respondents with LBP
### Prevalence
We report here (table 2) that weighted prevalence of LBP was 18.5% (95% CI: 11.8% to 25.2%). However, the age-standardised prevalence of LBP was reported to be 19.4% (95% CI: 14.0% to 24.8%), which is significantly higher in women (27.2%, 19.3% to 35.1%) than men (14.0%, 8.7% to 19.3%). There has been a persistent increase in prevalence from 10.5% (95% CI: 5.7% to 15.4%) in 18–34 years age group to 27.8% (95% CI: 16.1% to 39.5%) in 55–99 years age group. This trend was more prominent in women. The prevalence did not vary significantly among occupational groups. People with no formal education had significantly highest prevalence of LBP (31.3%, 22.3% to 40.4%) compared with other educational groups. Although the highest prevalence (23.5%, 13.9% to 33.0%) was observed in the first quartile of the wealth index, it did not vary significantly. LBP was not significantly associated with strenuous physical activity in our sample. We checked LBP prevalence by urban (14.6%, 9.9% to 19.2%) and rural (20.2%, 11.06% to 29.3%) categories, but it did not differ significantly. Among the comorbidities, the prevalence of LBP was higher among patients of hypertension (26.7%, 15.0% to 38.4%) and obesity (20.6%, 13.0% to 28.3%). The highest prevalence of LBP (87.3%, 80.2% to 94.4%) was seen in respondents who had multiple (two or more) MSDs such as LBP, knee osteoarthritis, soft tissue rheumatism, non-inflammatory MSDs, cervical spondylosis and so on (figure 2).

### Background characteristics
Among the LBP respondents (n=343), 63.3% were women, and 65.3% were from rural areas. Mean age in years (95% CI) was 45.3 (43.0 to 47.7) overall, and 48.3 (45.8 to 50.9) in men and 44.0 (41.0 to 47.0) in women. The study participants were divided into three subgroups as per age, and the highest number of LBP was observed in the 35–54 age group. More than half (%, 95% CI: 57.4%, 48.2% to 66.6%) were homemakers (all women), while the rest constituted other occupations like labourer, business professional, service holder and others. Almost half of the participants with LBP had no formal education (53.2%, 41.6% to 64.9%). Overall, according to the wealth index, 33.2% (22.6% to 43.9%) of respondents belonged to the first quartile (lowest socioeconomic status). About three-fourth of the respondents (77%, 55.9% to 98.0%) had rural residence (table 3).

### Disability and work loss
The distribution of Bengali HAQ-DI scores was not normally distributed among the patients with LBP. The scores ranged from 0 to 2.6. The LBP group's median (IQR) was 0.6 (0.3–0.9). The difference of Bangla HAQ-DI score between LBP (n=343) and no MSD (n=60) is statistically significant (p<0.0001) by the Kruskal-Wallis H test, indicating that LBP is associated with a higher disability. The distribution of days lost from work for LBP group had a highly skewed distribution and ranged from 0 to 365 days. However, the

**Table 2** Weighted prevalence of low back pain by sociodemographic characteristics in Bangladesh, musculoskeletal disease survey 2015

| Sociodemographic characteristics | Total<br>Per cent (95% CI) | Men<br>Per cent (95% CI) | Women |
|---|---|---|---|
| Overall | 18.5 (11.8 to 25.2) | 13.1 (6.4 to 19.9) | 22.7 (15.3 to 30.2) |
| Overall (age-standardised)* | 19.4 (14.0 to 24.8) | 14.0 (8.7 to 19.3) | **27.2 (19.3 to 35.1)** |
| Age in years | | | |
| 18–34 | 10.5 (5.7 to 15.4) | 5.2 (1.3 to 9.0) | 13.5 (7.2 to 19.9) |
| 35–54 | 23.1 (15.3 to 30.9) | 18.8 (8.7 to 28.8) | 26.7 (18.3 to 35.1) |
| 55–99 | 27.8 (16.1 to 39.5) | 15.7 (6.7 to 24.7) | 44.5 (23.9 to 65.1) |
| Occupation | | | |
| Homemaker | 23.6 (15.9 to 31.2) | ** | 23.6 (15.9 to 31.2) |
| Labourer† | 18.2 (9.4 to 27.0) | 17.9 (8.6 to 27.3) | 21.2 (6.2 to 36.3) |
| Business professional | 9.6 (1.9 to 17.2) | 9.8 (1.9 to 17.6) | ** |
| Service holder | 10.7 (1.2 to 20.1) | 10.3 (0.4 to 20.2) | ** |
| Others† | 13.9 (4.1 to 23.6) | 8.6 (1.7 to 15.5) | 20.3 (6.0 to 34.6) |
| Education | | | |
| No formal education (0) | 31.3 (22.3 to 40.4) | 20.3 (13.4 to 27.2) | 37.4 (24.3 to 50.5) |
| Any primary education (1–5) | 13.3 (6.7 to 19.9) | 12.1 (3.9 to 20.3) | 14.4 (7.8 to 21.0) |
| Any secondary education (6–10) | 14.9 (7.4 to 22.3) | 12.0 (2.4 to 21.6) | 17.5 (9.9 to 25.0) |
| Above secondary (≥11 years) | 6.9 (3.8 to 10.1) | ** | 8.0 (3.7 to 12.3) |
| Married‡ | 19.6 (12.4 to 26.7) | 14.1 (6.7 to 21.4) | 23.7 (15.7 to 31.7) |
| Wealth index quartile§ | | | |
| First | 23.5 (13.9 to 33.0) | 19.5 (11.2 to 27.8) | 25.5 (12.9 to 38.2) |
| Second | 19.8 (8.4 to 31.2) | 17.3 (3.7 to 30.9) | 21.6 (10.3 to 32.9) |
| Third | 14.1 (7.6 to 20.5) | 8.3 (2.3 to 14.2) | 19.8 (11.7 to 28.0) |
| Fourth | 16.6 (9.3 to 23.8) | 10.3 (3.2 to 17.4) | 23.1 (11.5 to 34.7) |
| Residence | | | |
| Urban | 14.6 (9.1 to 20.0) | 9.8 (3.9 to 15.6) | 17.7 (8.5 to 26.8) |
| Rural | 20.2 (10.5 to 29.8) | 14.3 (4.8 to 23.9) | 25.1 (14.7 to 35.4) |
| Strenuous physical activity¶ | 17.1 (4.3 to 29.8) | 17.2 (4.0 to 30.4) | ** |
| History of physical trauma | 24.5 (14.2 to 34.7) | 24.6 (13.0 to 36.2) | 24.4 (7.9 to 40.9) |

All values are per cent (95% CI). Weighted percentages shown are calculated from Census 2001 Population Frame by Bangladesh Bureau of Statistics to reflect projected population of Bangladesh (N=94 794 164).
Bold face values denote statistically significant higher prevalence in women compared with men.
Other occupations: retired, weaver and housekeeper.
*Standardised for WHO World Population 2000–2020.
†Labourer includes: farmer, daily worker, rickshaw puller, garments worker, field worker and others.
‡Includes currently married, divorced, separated and widowed.
§The wealth index was constructed using principal component analysis out of a list of 20 household assets (see Methods section for details);.
¶Fourth quartile of the metabolic equivalent tasks-minutes distribution of work-related physical activity. Commutation and leisure time physical activities were not considered because these were negligible contributors.
**95% CI not reported as number of respondents are <25.

difference of days lost from work (n=1625) between LBP (n=343) and no MSD (n=1282) is statistically significant (p<0.0001) by the Kruskal-Wallis H test, indicating that LBP is associated with more days lost.

### Factors associated

Univariate logistic regression analysis did not show any significant association of LBP with occupation, strenuous physical activity, wealth indices, tobacco use, obesity and diabetes (table 4). A significant association was observed for age group 35–99 years, female sex, lack of education, history of physical trauma and hypertension according to the unadjusted OR and their 95% CIs (p<0.05). These significant associations (p<0.01) persisted in the multiple logistic regression analysis having age, sex, education and hypertension into the model simultaneously.

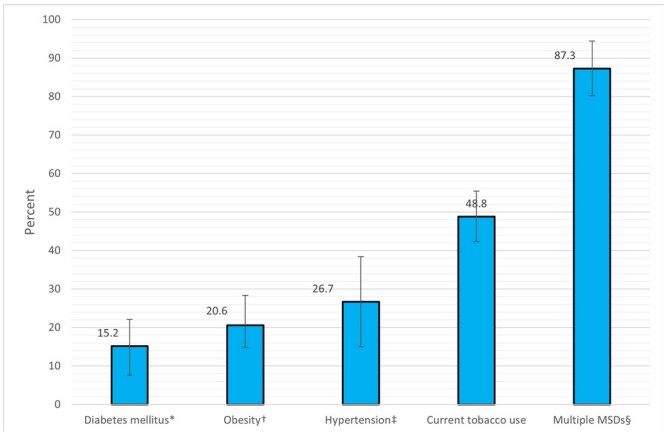

**Figure 2** Weighted prevalences (percent) of low back pain in comorbid conditions (error bars indicate 95% CIs). *Diabetes was defined as random capillary glucose level ≥11.1 or on medication for diabetes. †Obesity is defined as, body mass index ≥25 kg/m$^2$; ‡Hypertension was defined as systolic blood pressure ≥140 or diastolic blood pressure ≥90 or medication for hypertension; §Respondents with multiple musculoskeletal disorders (MSDs) suferred from low back pain, knee osteoarthritis, soft tissue rheumatism, non-inflammatory MSDs, cervical spondylosis etc.

## DISCUSSION

LBP is a common medical problem with very high personal and societal impact, leading to poor quality of life and workability.[31] In this survey, we report that one in five adults in Bangladesh suffers from LBP. The prevalence in Bangladesh is similar to Kuwait (22.7%)[32] and Qatar (23.3%)[33] but lower than northern India (32%)[17] and Iran (29.3%).[14] Malaysia (11.6%)[34] has a lower prevalence of LBP than we report here. In some previous studies in Bangladesh the prevalence was found to be 25.6% among medical students,[35] 36.6% in bank employees[36] and 60.8% among physiotherapists.[37] Lack of maintenance of correct posture during sitting and scarcity of knowledge, understanding or application of ergonomics are responsible for high prevalence rate among these groups.[35 37]

A systematic review of 165 studies from 54 countries revealed the global prevalence of LBP of 12–33%.[1] According to the systematic analysis of the Global Burden of Disease Study 2017, LBP was the leading cause of years lived with disability.[38] In our study, the rural people had a higher prevalence of LBP than the urban people. This difference–though not statistically significant–was probably due to lower doctor concentration in the rural areas, financial limitations and less education status. However, regional variation was observed in other studies.[14]

Age was a factor associated with LBP in this study. The prevalence of LBP persistently increased with age but this was not statistically significant. However, an analysis reported that the risk and prevalence of LBP increased with age.[5] A systematic review of the global prevalence of LBP revealed the association of age was highest in the 40–49 age group.[1 39] The overall prevalence rises with age

65, which gradually reduces thereafter.[40] Some possible explanations are LBP characteristics in older adults that differ from the middle-aged population (less intense back pain, more leg pain and more depression).[41] Our study showed a higher prevalence of LBP among women than men, consistent with some other analyses.[1 40] This could be due to more household or domestic activities among women compared with men. This finding agrees with the results from the national health survey on the Iranian population.[14] Another Indian study found no significant difference in age and sex scores in their study.[42] Higher women prevalence can be partially explained that they have a lower pain threshold than men.[43] The sex differences may be implied with gonadal steroid hormones like estradiol and testosterone that modulate sensitivity to pain and analgesia.[44] Women always experienced a higher frequency of visceral pain (eg, during menstruation, pregnancy) than men.[45] It seems that in painful conditions, women exhibit a greater prevalence than men as women report more pain.[46]

Leboeuf-Yde considered body weight as a possible weak indicator for LBP in his systematic literature review due to lack of evidence.[47] The results obtained in our study did not demonstrate a statistically significant association between LBP and higher BMI. Some other studies found obesity or high BMI associated with increased risk of LBP development and severity.[14 48] However, a cross-sectional study including nine countries found BMI ≥25 kg/m$^2$ as a risk factor for LBP in five countries (Finland, Poland, Russia, South Africa and Spain), whereas it was not associated with LBP in the remaining four countries (China, Ghana, India and Mexico).[49]

In our research, we found that the absence of formal education is significantly associated with LBP. A cross-sectional study in the USA found that LBP is more common in people who have had less than high school education.[50] Other studies in the USA, UK and Iran found lower educational status had an increased association with LBP and found higher education inversely associated with LBP.[3 14] Several proposed mechanisms may account for the relationship between low academic status and back pain. The amount of formal education contributes to the types of jobs that an individual may involve in, and subsequently, the types of jobs that influence LBP.[51] Moreover, health education regarding posture management, lifestyle changes, physical exercises, stress management poorly reached among people with an absence of formal education. We did not find any significant association of LBP with occupation. However, in our opinion, the larger number of homemakers affected with LBP might be linked with their nature of heavy work such as squatting, bending lifting heavy objects, prolonged standing and so on in the household. In a US-based study, LBP was significantly related to occupational factors such as truck driving, lifting, carrying, pulling, pushing, twisting and non-driving vibrational exposure.[52] In some European countries, workers involved in heavy weightlifting (≥25 kg) suffered more from LBP.[53]

**Table 3** Socioeconomic characteristics of patients of low back pain in Bangladesh, musculoskeletal disease survey 2015

| Sociodemographic characteristics | Total | Men | Women |
|---|---|---|---|
| | Weighted percentage (95% CI) | | |
| Age in years | | | |
| Mean (95% CI) | 45.3 (43.0 to 47.7) | 48.3 (45.8 to 50.9) | 44.0 (41.0 to 47.0) |
| 18–34 | 24.8 (18.8 to 30.7) | 14.1 (6.9 to 21.3) | 29.6 (21.3 to 37.9) |
| 35–54 | 46.4 (40.6 to 52.2) | 55.6 (40.3 to 70.9) | 42.3 (34.4 to 50.2) |
| 55–99 | 28.8 (22.5 to 35.1) | 30.3 (17.0 to 43.7) | 28.1 (22.2 to 34.0) |
| Occupation | | | |
| Homemaker | 57.4 (48.2 to 66.6) | – | 83.2 (72.0 to 94.5) |
| Labourer* | 20.4 (13.6 to 27.2) | 59.7 (38.8 to 80.5) | 2.7 (0.1 to 5.4) |
| Business professional | 5.2 (1.2 to 9.1) | 16.6 (5.0 to 28.2) | – |
| Service holder | – | – | – |
| Others* | 13.5 (2.7 to 24.3) | 14.8 (2.8 to 26.8) | 13.0 (1.5 to 24.4) |
| Education | | | |
| No formal education (0) | 53.2 (41.6 to 64.9) | 39.1 (24.3 to 54.0) | 59.6 (46.9 to 72.3) |
| Any primary education (1–5) | 17.2 (13.8 to 20.5) | 23.9 (17.1 to 42.6) | 14.1 (10.1 to 18.1) |
| Any secondary education (6–10) | 24.1 (15.9 to 32.4) | 29.9 (17.1 to 42.6) | 21.6 (13.1 to 30.1) |
| Above secondary (≥11 years) | 5.5 (1.0 to 9.9) | – | – |
| Married‡ | 97.8 (94.8 to 100.8) | 96.8 (92.2 to 101.3) | 98.3 (95.5 to 101.1) |
| Wealth index quartile§ | | | |
| First | 33.2 (22.6 to 43.9) | 30.1 (18.8 to 41.4) | 34.7 (21.7 to 47.7) |
| Second | 25.5 (14.8 to 36.2) | 29.5 (11.5 to 47.5) | 23.7 (14.6 to 32.7) |
| Third | 19.7 (11.0 to 28.4) | 18.5 (6.7 to 30.3) | 20.3 (10.5 to 30.1) |
| Fourth | 21.6 (9.2 to 33.9) | 21.9 (9.4 to 34.5) | 21.4 (7.1 to 35.7) |
| Rural residence | 77.0 (55.9 to 98.0) | 80.3 (60.2 to 100.4) | 75.5 (52.8 to 98.1) |

*Labourer include: farmer, daily worker, rickshaw puller, garments worker, field worker and others. Other occupations: retired, weaver and housekeeper, and so on.
†Numbers are low.
‡Includes currently married, divorced, separated and widowed.
§The wealth index was constructed using principal component analysis out of a list of 20 household assets (see Methods section for details);.
¶All values are per cent (95% CI) unless stated otherwise. Weighted percentages shown are calculated from Census 2001 Population Frame by Bangladesh Bureau of Statistics to reflect projected population of Bangladesh (n=94 794 164).

Studies reported associations between LBP and lower social class[14 50] but we did not find any significant association of LBP with economic status. This finding is coherent with a Danish study where they did not find any possible relationship between socioeconomic status and LBP.[54] In our analysis, trauma tended to be associated (unadjusted OR) with overall LBP, but the association was lost after adjustment. Trauma is not supposed to lead to chronic LBP and the persistence of back pain was more associated with psychological factors like stress, low education status, and so on, than trauma itself.[55] We found a positive relationship between hypertension and LBP. LBP may cause inactivity and lack of exercise resulting in weight gain, subsequently creating or exaggerating comorbid conditions like hypertension (HTN) and diabetes mellitus. The Hong Kong Disc Degeneration-Cardiovascular Cohort showed that HTN increases the possibility of moderate or severe disc degeneration which is highly associated with LBP.[56] Another long-term Finnish study revealed that both systolic blood pressure (SBP) and diastolic blood pressure (DBP) were positively associated with LBP suggesting atherosclerosis of lumbar vessels a possible mechanism of development of LBP.[57] In a Korean survey, the lifetime prevalence of LBP was 34.4% among the hypertensive respondents, but the adjusted OR of LBP prevalence was significantly lower than the normotensive subjects (fully adjusted OR: 0.79, 95% CI: 0.70 to 0.90).[58] A Bangladeshi study conducted among the doctors working in a tertiary care hospital found that HTN was the most common comorbid condition among the LBP sufferers.[59] There was no association between diabetes and LBP in this study.

### Strengths and limitations
This is the first nationally representative survey report on LBP in Bangladesh, and probably, among all south Asian

**Table 4** ORs of factors associated with LBP compared with no MSDs in Bangladeshi adults, musculoskeletal disease survey 2015

| Factors | Odds ratio (95% CI) | |
| --- | --- | --- |
| | **Unadjusted** | **Adjusted** |
| Age group, years | 3.2 (2.5 to 4.2)** | 2.4 (1.7 to 3.4)** |
| (35–99=1, 18–34=0) | 1.0 | 1.0 |
| Sex | 2.1 (1.3 to 3.3)** | 2.2 (1.5 to 3.3)** |
| (woman=1, man=0) | 1.0 | 1.0 |
| Labourer† | 0.9 (0.7 to 1.3) | – |
| (yes=1, no=0) | 1.0 | – |
| No formal education | 3.5 (2.5 to 5.0)** | 2.3 (1.6 to 3.3)** |
| (yes=1, no=0) | 1.0 | 1.0 |
| Low wealth index | 1.6 (1.1 to 2.3)** | 1.0 (0.6 to 1.5) |
| (yes=1, no=0) | 1.0 | 1.0 |
| Strenuous physical activity‡ | 0.8 (0.4 to 1.6) | – |
| (yes=1, no=0) | 1.0 | – |
| Obesity (body mass index ≥25 kg/m$^2$) | 1.3 (0.8 to 2.1) | – |
| (yes=1, no=0) | 1.0 | – |
| History of physical trauma | 1.8 (1.1 to 3.2)* | 1.6 (0.9 to 2.8) |
| (yes=1, no=0) | 1.0 | 1.0 |
| Current tobacco user | 1.1 (0.8 to 1.6) | – |
| (yes=1, no=0) | 1.0 | – |
| Hypertension | 2.3 (1.3 to 4.0)** | 1.7 (1.1 to 2.6)* |
| (yes=1, no=0) | 1.0 | 1.0 |
| Diabetes mellitus | 1.0 (0.5 to 1.7) | – |
| (yes=1, no=0) | 1.0 | – |

*p<0.05, **p<0.01.
†Labourer includes: farmer, daily worker, rickshaw puller, garments worker, field worker and others.
‡Fourth quartile of the metabolic equivalent tasks-minutes distribution of work-related physical activity. Commutation and leisure time physical activities were not considered because these were negligible contributors.
LBP, low back pain; MSD, musculoskeletal disorder.

nations. Although we have weighted the data for national representation, the sample size calculation for the original study was based on point prevalence of MSD.[19] We now know that the prevalence of LBP was 19.4% and prevalence of MSD was 30.4%. A larger sample size maintaining adequate power was needed for generalisability of the study results. Therefore, a cautious interpretation is necessary because of inadequate sample size, especially when split into reporting domains. We have operationally defined the recall period for reporting work loss days as 12 months which might induce bias. Trained rheumatology residents diagnosed the patients which was verified by experienced rheumatologists in the field. Some diagnoses of evolving rheumatological conditions might lack sufficient validity because of a lack of adequate laboratory facilities in the field.

## CONCLUSION

This nationally representative study reports the population weighted prevalence of LBP by sociodemographic background, comorbidities and associated factors in Bangladesh. One in five adults suffer from LBP. Education and hypertension are modifiable factors that warrant intervention. Increase in level of education, care to middle and older population and good control of hypertension may reduce LBP burden. A special attention is needed to prevent LBP in women. Further study with a larger sample size addressing these neglected issues may have more clarifications to decrease the burden of LBP.

**Acknowledgements** We gratefully acknowledge the field team, divisional coordinators, civil surgeons, Upazila health and family planning officers and health assistants for their support.

**Contributors** MSMM: conceptualised the area of work, prepared the first draft of the manuscript and revised the subsequent drafts and is the guarantor of the overall content. FH: analysed the data, guided preparation of graphs and tables, interpreted the results, revised the draft manuscript and coordinated the writing exercise. IHB: prepared graphs and tables and revised the draft manuscript. MAR:

reviewed the manuscript and with special reference to the literature review. AZ-A-Q and SAH: prepared the training manual, trained the field team, executed the field operation in coordination with the divisional investigators. MRC, SAH and MMZ: designed the study, guided the analysis and manuscript writing, critically reviewed the results and manuscript drafts and ensured integrity of data. All authors approved publication.

**Funding** Preparing this manuscript did not require any funding. However, the base study was supported by the World Health Organization, Bangladesh (Agreement Reference: SEBAN140895) back in 2015. As a part of its mandate to strengthen national research capacity, WHO provided technical guidance in designing, implementing, analysing data and writing the report. However, it does not have any influence on the results.

**Competing interests** None declared.

**Patient and public involvement** Patients and/or the public were not involved in the design, or conduct, or reporting, or dissemination plans of this research.

**Patient consent for publication** Not applicable.

**Ethics approval** This study involves human participants and was approved by Ethical clearance was obtained from the Institutional Review Board of Bangabandhu Sheikh Mujib Medical University (ID 1100). Participants gave informed consent to participate in the study before taking part.

**Provenance and peer review** Not commissioned; externally peer reviewed.

**Data availability statement** Data are available upon reasonable request. Yes, I agree that all the free text of the manuscript be published.

**ORCID iDs**
Muhammad Shoaib Momen Majumder http://orcid.org/0000-0002-7901-0847
Ferdous Hakim http://orcid.org/0000-0003-2376-3978
Iftekhar Hussain Bandhan http://orcid.org/0000-0002-3800-6155
Minhaj Rahim Choudhury http://orcid.org/0000-0002-8695-5240
MM Zaman http://orcid.org/0000-0002-1736-1342

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
