## [Reviewer comments · BMJ Open]

ARTICLE DETAILS

TITLE (PROVISIONAL)	Low Back Pain in the Bangladeshi Adult Population: A Cross-sectional National Survey
AUTHORS	Momen Majumder, Muhammad Shoaib; Hakim, Ferdous; Bandhan, Iftekhar; Razzaque, Mohammad Abdur; Zahid-Al-Quadir, Ahmad; Ahmed, Shamim; Choudhury, Minhaj; Haq, Syed; Zaman, MM

VERSION 1 – REVIEW

REVIEWER	Shabbir Sany Faridpur Medical College, Community Medicine
REVIEW RETURNED	06-Dec-2021

GENERAL COMMENTS	Thank you for giving me the opportunity to review this interesting manuscript. This cross-sectional study determined the residence- and sex-specific prevalence and risk factors of LBP among Bangladeshi people. Overall, it is a well-designed study, and the methodology was described step to step. However, there are a few issues, especially references, that deserve attention. I have made some recommendations, and I hope it will help enhance the quality of this manuscript. The reviewer provided a marked copy with additional comments. Please contact the publisher for full details.
---

REVIEWER	Nasima Akter Fouzder Hat Nursing College
REVIEW RETURNED	22-Dec-2021

GENERAL COMMENTS	thanks for a good research
----------------------------

REVIEWER	Aleksander Galaś Jagiellonian University, Department of Epidemiology and Preventive Medicine
REVIEW RETURNED	09-Mar-2022

GENERAL COMMENTS	Dear Editor, Dear Authors, Thank You for allowing me to provide some comments on the manuscript “Low Back Pain in the Bangladeshi Adult Population: A Cross-sectional National Survey”. The topic is valuable, as there is limited data about LBP in that region. The manuscript, however, suffers from several limitations listed below, which, in my opinion, should be considered before publication.
--

	Key messages: -I suggest referring to some value instead of “a common”; global prevalence of LBP was not investigated in this study!-the design of the presented study does not allow to state on risk factors, which implies causality-under the “impact on future clinical practice” – the sentence provided does not address this topic The purpose of the study is expressed too generally. The authors should say clearly what was the aim instead of saying “This communication focuses more deeply on the ...” If the purpose was to show the prevalence of LBP in Bangladesh, as the authors applied (if I understand it correctly) a household level multi-stage stratified sampling, they should provide population weighted data. To provide more valuable information the Authors should provide weighted prevalence to provide descriptive data for the LBP in Bangladesh. Results and conclusions should be based on the weighted data. As the study design does not allow to state whether factors associated with LBP are really the risk factors I suggest avoiding calling these factors as ‘risk factors’. Some of them, like education, are just proxy measures. It should be clarified under Methods what is age-adjusted prevalence. The weighting procedure should be added and clearly described. I try to avoid guessing, but as the objective of the study is not clearly defined to me it is hard to say whether study design was appropriate. Surely, to determine risk factors for LBP the design of cross-sectional study is not appropriate. Authors should provide, at least in the supplementary materials, or under methods, the whole step-by-step strategy (questions, examinations) which finally led to identification of (to be diagnosed with) LBP. Ethics: it is hard to say, whether an informed consent was obtained from each participant, as it is not clearly stated. Authors discuss the association between LBP and obesity, that is quite good, but in their discussion they omitted some key publications in the area, like Koyanagi et al. The association between obesity and back pain in nine countries: a cross-sectional study which included >42.000 individuals. Results: I suggest presenting firstly the frequency of LBP in the study group un-weighted (currently Tab.2) & weighted and next the characteristic of LBP (currently Tab.1). The limitations should be discussed more broadly. Conclusions: A comment that it is “the first study to report” is not a conclusion. Calling the associated factors as risk factors is not a conclusion supported by the results of this study. And additionally, referring to appropriate health education and clinical services is very general and was not investigated in this study either. I suggest providing the STROBE checklist for cross-sectional studies. Best regards, Reviewer.
--	---

VERSION 1 – AUTHOR RESPONSE

Reviewer 1: Dr. Shabbir Sany	
1.	References missing: Page 4, line 114 Page 4, line 121, 124 Page 4, line 132
	We have inserted references for the statements as identified. Reference No. 1, page 4, line 118 Reference No. 7, page 4, lines 125, 130 Reference No. 4, page 4, line 137 ¹ Hoy D, Bain C, Williams G, et al. A systematic review of the global prevalence of low back pain. Arthritis Rheum. 2012 Jun;64(6):2028-37. doi: 10.1002/art.34347. ⁴ Balagué F, Mannion AF, Pellisé F, et al. Non-specific low back pain. Lancet. 2012 Feb 4;379(9814):482-91. doi: 10.1016/S0140-6736(11)60610-7. ⁷ Hartvigsen J, Hancock MJ, Kongsted A, et al. Lancet Low Back Pain Series Working Group. What low back pain is and why we need to pay attention. Lancet. 2018 9 June;391(10137):2356-2367. doi: 10.1016/S0140-6736(18)30480-X.
2.	What did you mean by 'unconventional way'? Could you please elaborate briefly? (page 4, lines 124)
	We have revised the text as, “Disability from LBP is a primary concern for the LMICs including Bangladesh where manual labour– rickshaw pulling, day labourers, housemaids, work exposures to the lifting of heavy weight during their day-to-day activities etc.– is common⁷. The scope for job switching is restricted in resource constraint countries.” (page 4, lines 127–130) ⁷ Hartvigsen J, Hancock MJ, Kongsted A, et al. Lancet Low Back Pain Series Working Group. What low back pain is and why we need to pay attention. Lancet. 2018 9 June;391(10137):2356-2367. doi: 10.1016/S0140-6736(18)30480-X.
3.	Please re-check the definition of non-specific low back pain provided by Balagué et al. Non-specific low back pain was defined as a symptom without any known cause rather than the
	We have revised the text as follows, “Non-specific LBP is defined as LBP not particularly associated with specific aetiology like malignancy, infection, fracture, inflammatory condition, radiculopathy or cauda equina syndrome⁴.” (page 4, line 137–138)

	absence of any particular condition. (page 4, lines 134)	4 Balagué F, Mannion AF, Pellisé F, et al. Non-specific low back pain. Lancet . 2012 Feb 4;379(9814):482-91. doi: 10.1016/S0140-6736(11)60610-7.
4.	The authors included the reference of only one study; references of other studies are missing. (page 4, line 137–138)	We have inserted references for the statement as identified, “The estimated lifetime prevalence was 84.1% in a Canadian study¹¹, 70% in Denmark¹², 59% in the UK¹³. In Iran, the prevalence of LBP was 29.3%¹⁴.” (page 4, lines 142–143) ¹¹ Cassidy JD, Carroll LJ, Côté P. The Saskatchewan health and back pain survey. The prevalence of low back pain and related disability in Saskatchewan adults. Spine. 1998 Sep 1;23(17):1860-6; discussion 1867. doi: 10.1097/00007632-199809010-00012. ¹² Harreby M, Kjer J, Hesselsøe G, Neergaard K. Epidemiological aspects and risk factors for low back pain in 38-year-old men and women: a 25-year prospective cohort study of 640 school children. Eur Spine J. 1996;5(5):312-8. doi: 10.1007/BF00304346. ¹³ Waxman R, Tennant A, Helliwell P. A prospective follow-up study of low back pain in the community. Spine. 2000 Aug 15;25(16):2085-90. doi: 10.1097/00007632-200008150-00013.
5.	Could you please confirm whether any reporting guideline (e.g., STROBE) was followed? (page 5, lines 149)	We have followed the STROBE checklist for cross-sectional studies as mentioned in reply to Editor Comment No. 3.
6.	Please describe the relevant dates of the study conducted and data collection (as per item number 5 of the STROBE checklist). Please, include the completed STROBE checklist as an external file and include the reference. (page 5, lines 149)	We have mentioned the dates of the study and data collection, “Data was collected in November and December of 2015.” (page 5, lines 167–168) We included the STROBE checklist for cross-sectional studies as mentioned in reply to Editor Comment No. 3.
7.	How validation was done? (page 5, lines 165)	We revised the text as follows, “The English version of the first part of the questionnaire was translated to Bangla, then adapted according to the guideline of Beaton et al. ²² , validated by Chassany’s Method ²³ , and

		administered by the interviewers.” (page 5, lines 173–175). ²² Beaton DE, Bombardier C, Guillemin F, Ferraz MB. Guidelines for the process of cross-cultural adaptation of self-report measures. Spine (Phila Pa 1976). 2000 Dec 15;25(24):3186-91. doi: 10.1097/00007632-200012150-00014. ²³ Chassany O, Marquis P, Scherrer B, Read NW, Finger T, Bergmann JF, Freitag B, Geneve J, Caulin C. Validation of a specific quality of life questionnaire for functional digestive disorders. Gut. 1999 Apr;44(4):527-33. doi: 10.1136/gut.44.4.527.
8.	What did you mean my ‘suspected’. What were the criteria to be regarded as ‘suspected respondent’? How did you confirm the diagnosis, and how did the investigator validate it? (page 5, lines 168, 169)	We have revised the text added as “A subject was considered a positive respondent if he/ she reported pain in muscles, bones, joints, or any part of the body (musculoskeletal system) during the preceding seven days. Subjects who were taking pain medications like non-steroidal anti-inflammatory drugs (NSAID) or steroids were considered positive respondents even if they did not report pain on those seven days. All positive respondents were interviewed and examined by the research physicians. Internationally accepted criteria were used for the diagnosis of the diseases. For conditions without internationally accepted criteria, relevant investigations and clinical judgment of the research physician were used. The final diagnoses were checked and verified by a rheumatologist during their visit to respective PSUs.” (page 5–6, lines 177–185).
9.	Please briefly describe how you measured the random blood glucose level, body mass index and waist circumference. (page 6, line 195)	Added text, “Random capillary blood glucose was measured by using glucometer (Accu-Check Germany). Using height (meters) and weight (kilograms) measurements, we calculated BMI [(weight (kg)/height (meter)²]. Waist circumference was measured by placing a measuring tape horizontally above the iliac crest. (page 6, lines 210–213)
10.	The words ‘obesity’ and ‘overweight’ were used interchangeably in the manuscript. However, the definition of these words is different. Please correct it. (page 6, line 196)	We have revised it by using the term ‘obesity’ throughout the manuscript to indicate BMI ≥ 25 kg/m² as per Asian Classification. “Obesity was defined as a body mass index of ≥ 25 kg/meter squared²⁷.” (page 6, lines 214–215) ²⁷ Misra A. Ethnic-Specific Criteria for Classification of

		Body Mass Index: A Perspective for Asian Indians and American Diabetes Association Position Statement. Diabetes Technol Ther. 2015 Sep;17(9):667-71. doi: 10.1089/dia.2015.0007.
11.	Please re-check reference number: 23 (page 6, line 197)	We have checked the reference and omitted it.
12.	How was the test of normality performed? (page 6, line 204)	We would humbly disagree with using a detailed description of the issue and consider it unnecessary. No changes are made.
13.	Consent – Oral, written or both? (page 7, line 218)	Written consent was obtained. (page 7, line 246–247)
14.	Please include the ‘P-value’ where applicable in the result section of the manuscript (exact P-value is preferable). (page 7, line 225)	‘P-value’ has been mentioned in the ‘results’ section as applicable: page 9 and 10, line 306, 310, 318, and 319. However, the section on ‘risk factors’ has P values in addition to 95% CIs for ease of understanding. The use of exact P values for all the variables will unnecessarily make the table clumsy. We presented 95% CIs, which is clinically more important than P values. Therefore, we are not changing it. No change is made to other parts of the manuscript that includes 95% confidence intervals (CI).
15.	Could you demonstrate the results in an additional table, please? (page 7, line 226)	These results are reported in detail in the article by Zahid-AI-Quadir et al.¹⁹. We have reiterated the pertinent results with reference to orient the reader on the characteristics of respondents. No changes are made to the manuscript. ¹⁹ Zahid-AI-Quadir, A., Zaman, M.M., Ahmed, S. et al. Prevalence of musculoskeletal conditions and related disabilities in Bangladeshi adults: a cross-sectional national survey. BMC Rheumatol 4, 69 (2020). https://doi.org/10.1186/s41927-020-00169-w
16.	The authors stated 52.2% were homemakers; however, table 1 shows 52.5% were homemakers. Please correct it.	The results have changed as we have reported the weighted percentages in response to Reviewer 3 Comment No. 4. Now the weighted result is 57.7%. We have ensured consistency of results cited in the main text with those presented in the tables.
17.	Age-adjusted prevalence of LBP in women is reported as (23.5%, 16.0–31.0), while in table 2, it is reported as 27.2% (19.3–35.1); please correct it.	We have revised the text as ‘age-standardization’ throughout the manuscript and in Table 2. “Age-standardization of prevalence estimates was made for global comparison using the WHO World Population 2000-2025 ²⁹ .” (page 7, line 234–235)

		²⁹ Ahmad OB, Boschi-Pinto C, Lopez AD, et al. Age Standardization of Rates: A New WHO Standard. GPE Discussion Paper Series: No31 Geneva, Switzerland: World Health Organization, 2001.
18.	The prevalence of LBP was 20.6% among the participants with BMI \geq 25 kg/m²; however, in figure 2, it showed 20.2%. Please correct it.	The weighted prevalence is 20.6%, which has been presented consistently everywhere.
19.	In discussion, the authors compared the study results with other study findings. However, there are some studies conducted on Bangladeshi people of different professions with LBP. May I suggest comparing the findings of this study with those studies? (page 9, line 287)	We have revised the text with references: "In some previous studies in Bangladesh, the prevalence was found 25.6% among medical students³⁶, 36.6% in bank employees³⁷, 60.8% among physiotherapists³⁸. Lack of maintenance of correct posture during sitting and scarcity of knowledge, understanding, or application of ergonomics are responsible for the high prevalence rate among these groups^{36 38}. (page 11, line 328-332) ³⁶ Sany SA, Tanjim T and Hossain MI. Low back pain and associated risk factors among medical students in Bangladesh: a cross-sectional study. F1000Research. 2021 July; 10:698. doi: 10.12688/f1000research.55151.1. ³⁷ Ali M, Ahsan GU, Hossain A. Prevalence and associated occupational factors of low back pain among the bank employees in Dhaka City. J Occup Health. 2020 Jan; 62(1):e12131. doi: 10.1002/1348-9585.12131. ³⁸ Mondal R, Sarker RC, Akter S, Banik PC, Baroi SK. Prevalence of low back pain and its associated factors among physiotherapists in Dhaka city of Bangladesh in 2016. Journal of Occupational Health and Epidemiology. 2018. 7:70–4. doi: 10.29252/johe.7.2.70.
20.	The authors compared the findings of the study with other study findings. However, in case of any discrepancies, they did not describe the possible reason. (page 9, line 287)	We have compared the results with other authors and provided reasons for agreement and disagreement except for socioeconomic status. We understand that not all risk factors are universally applicable, and country or population-specific factors may play a role in the development of LBP. (page 10; 11; 12; lines 330–332, 342–3464; 354–358, 362–369; 390–392, 412–

		415.
21.	References of 'other analyses' are missing. (page 9, line 308)	We have added the relevant references (Ref. No. 1 and 41). (page 11, lines 354–356) ¹ Hoy D, Bain C, Williams G, et al. A systematic review of the global prevalence of low back pain. Arthritis Rheum. 2012 Jun;64(6):2028-37. doi: 10.1002/art.34347. ⁴¹ Loney PL, Stratford PW. The prevalence of low back pain in adults: a methodological review of the literature. Phys Ther. 1999 Apr;79(4):384-96. PMID: 10201544.
22.	What could be the possible reason and mechanism of significant association between LBP and trauma? Please describe it briefly. Also, compare this finding with other studies. (page 10, lines 341)	The weighted result did not find any significant relationship between trauma with LBP. We have revised the text as follows, “In our analysis, trauma tended to be associated (unadjusted OR) with overall LBP, but the association was lost after adjustment. Trauma is not supposed to lead to chronic LBP, and the persistence of back pain was more associated with psychological factors like stress, low education status etc., than trauma itself⁵⁷.” (page 12, lines 400–412) ⁵⁷ Harris IA, Young JM, Rae H, et al. Factors associated with back pain after physical injury: a survey of consecutive major trauma patients. Spine (Phila Pa 1976). 2007 15 June;32(14):1561-5. doi: 10.1097/BRS.0b013e318067dce8.
23.	'For determining work loss, the recall period was 12 months' – The authors did not mention the possible recall bias. (page 11, lines 354)	The revised text under 'strengths and limitations to include this unintentional omission. (page 3, 13 line 96, 434–435.
24.	Generalisability: As per item number 21 of the STROBE checklist - Discuss the generalisability (external validity) of the study results. (page 11, lines 354)	We cannot ensure 100% generalizability of the study results as a larger sample size would have carried more power. We have revised the text as, “Although we have weighted the data for national representation, the sample size calculation for the original study was based on point prevalence of MSD19. We now know that the prevalence of LBP was 18.5%, and the prevalence of MSD was 30.4%. A larger sample size maintaining adequate power was needed for the generalizability of the study results. Therefore, a cautious interpretation of the results is necessary because of inadequate sample size, especially when

		split into reporting domains.” (page 13, lines 428–433)
25.	The authors addressed the risk factors; however, the potential solutions could have been suggested, especially for modifiable risk factors. (page 11, lines 365)	We revised the text as, “Increase in the level of education, care to the middle and older population, and good control of hypertension may reduce LBP burden. Special attention is needed to prevent LBP in women. Further study with a larger sample size addressing these neglected issues may have more clarifications to decrease the burden of LBP.” (page 13, lines 446–449).
26.	Recommendations for future research could have been added.	We have revised the text in that line as mentioned above response.
Reviewer 2: Dr. Nasima Akter		
	Thanks for a good research	Thank you for the valuable review and compliment.
Reviewer 3: Dr. Aleksander Gafás		
1.	Key messages: I suggest referring to some value instead of “a common”; global prevalence of LBP was not investigated in this study!	The section on ‘key messages’ has been replaced by ‘Strengths and Limitations’.
2.	Key messages: The design of the presented study does not allow to state on risk factors, which implies causality - under the “impact on future clinical practice” – the sentence provided does not address this topic	Please see the above response.
3.	The purpose of the study is expressed too generally. The authors should say clearly what was the aim instead of saying “This communication focuses more deeply on the ...” (page 4–5, line 145–147)	The purpose of the study is rephrased as, “We have further analyzed the data from the 2015 study and report the population-weighted prevalence according to sociodemographic factors, comorbid conditions, disability and work loss due to LBP, and identify the risk factors of LBP.” (page 5, lines 151–154)
4.	If the purpose was to show the prevalence of LBP in Bangladesh, as the authors applied (if I understand it correctly) a household level multi-stage stratified sampling, they should provide population weighted data. To provide more valuable information the Authors should provide weighted prevalence to	We have weighted the data and prepared the manuscript to show weighted results. We have added this text in the methods section “We have weighted ²⁸ the data to reflect the population frame of Bangladesh for the year 2015. Base weight– for the sampled population–was calculated using the probability of selection of respondents among the eligible number of members of the household in a cluster defined by division (37), age groups (3) and sex (2). The base weight was adjusted with non-response weights

	provide descriptive data for the LBP in Bangladesh. Results and conclusions should be based on the weighted data.	separately for males and females. The final weight was generated after calibration to frame population (2015) in domains by division, sex and age groups. Analysis was done using the final weights.” (page 7, lines 217–224) ²⁸ Hakim, F., Bhuiyan, R., Akter, M. K., Mohit, M. A., Alam, M. F., Karim, M. R., & Zaman, M. M. (2022). Weighting National Survey Data in Bangladesh: Why, How and Which weight? Weighting survey data. Bangladesh Medical Research Council Bulletin, 47(2), 118–126. https://doi.org/10.3329/bmrcb.v47i2.57769
5.	As the study design does not allow to state whether factors associated with LBP are really the risk factors. I suggest avoiding calling these factors as ‘risk factors’. Some of them, like education, are just proxy measures.	We have used the term as is used by many colleagues. References used are mentioned below. No change is made to the manuscript. ⁴³ Ganesan S, Acharya AS, Chauhan R, Acharya S. Prevalence and risk factors for low back pain in 1,355 young adults: a cross-sectional study. Asian spine journal. 2017 Aug;11(4):610. ⁴⁰ Hoy D, Brooks P, Blyth F, Buchbinder R. The epidemiology of low back pain. Best practice & research Clinical rheumatology. 2010 Dec 1;24(6):769-81.
6.	It should be clarified under Methods what is age-adjusted prevalence. The weighting procedure should be added and clearly described.	We have revised the word ‘age-adjusted’ to ‘age-standardization’ to avoid redundancy, as “Age-standardization of prevalence estimates was made for global comparison using the WHO World Population 2000-2025 (page 7, line 234–235) The weighting procedure is detailed as replied to Reviewer 3 Comment No. 4.
7.	I try to avoid guessing, but as the objective of the study is not clearly defined to me it is hard to say whether study design was appropriate. Surely, to determine risk factors for LBP the design of cross-sectional study is not appropriate.	Kindly see our response to Reviewer 3 Comment No. 3 above. Using cross-sectional studies to describe risk factors is common in epidemiological studies. Therefore, we humbly do not make any change.
8.	Authors should provide, at least in the supplementary materials, or	We have added a brief step-by-step description on the diagnosis of LBP “A subject was considered a positive

	under methods, the whole step-by-step strategy (questions, examinations) which finally led to identification of (to be diagnosed with) LBP.	respondent if he/she reported pain in muscles, bones, joints, or any part of the body (musculoskeletal system) during the preceding seven days. Subjects who were taking pain medications like non-steroidal anti-inflammatory drugs (NSAID) or steroids were considered positive respondents even if they did not report pain on those seven days. All positive respondents were interviewed and examined by the research physicians. Internationally accepted criteria were used for the diagnosis of the diseases. For the conditions without any internationally accepted criteria, relevant investigations and clinical judgment of the research physician were used. A rheumatologist checked and verified the final diagnoses during their visit to respective PSUs.” on pages 5–6, lines 177–185.
9.	Ethics: it is hard to say, whether an informed consent was obtained from each participant, as it is not clearly stated.	“Informed written consent was obtained from the study participants.” (page 7, line 246–247)
10.	Authors discuss the association between LBP and obesity, that is quite good, but in their discussion, they omitted some key publications in the area, like Koyanagi et al. The association between obesity and back pain in nine countries: a cross-sectional study which included >42.000 individuals.	We have revised the text in the discussion section incorporating an additional reference as suggested, “However, a cross-sectional study including nine countries found BMI $\geq 25\text{kg/m}^2$ as a risk factor for LBP in five countries (Finland, Poland, Russia, South Africa and Spain), whereas it was not associated with LBP in the remaining four countries (China, Ghana, India and Mexico) ⁵⁰ .” (page 11, lines 375–378) ⁵⁰ Koyanagi A, Stickley A, Garin N, Miret M, Ayuso-Mateos JL, Leonardi M, Koskinen S, Galas A, Haro JM. The association between obesity and back pain in nine countries: a cross-sectional study. BMC Public Health. 2015 Feb 11;15:123. doi: 10.1186/s12889-015-1362-9.
11.	Results: I suggest presenting firstly the frequency of LBP in the study group un-weighted (currently Tab.2) & weighted and next the characteristic of LBP (currently Tab.1).	We have rearranged the results section as per the suggestion as follows, Table 1: Characteristics of the respondents with LBP by unweighted and weighted numbers, Musculoskeletal Disease Survey 2015 Table 2: Weighted prevalence of low back pain by sociodemographic characteristics in Bangladesh, Musculoskeletal Disease Survey 2015 Table 3: Socioeconomic characteristics of patients of low back pain in Bangladesh, Musculoskeletal Disease Survey 2015 Table 4: Odds ratios of risk factors for low back pain

		compared with no musculoskeletal disorders in Bangladeshi adults, Musculoskeletal Disease Survey 2015 (page 17–21)
12.	The limitations should be discussed more broadly.	Changes made are mentioned in reply to Editor Comment No. 4.
13.	Conclusions: A comment that it is “the first study to report” is not a conclusion. Calling the associated factors as risk factors is not a conclusion supported by the results of this study. And additionally, referring to appropriate health education and clinical services is very general and was not investigated in this study either.	Revised accordingly, “This nationally representative study reports the population-weighted prevalence of LBP by sociodemographic background, comorbidities and risk factors in Bangladesh. One in five adults suffers from LBP. Education and hypertension are modifiable risk factors that warrant intervention. An increase in the level of education, care for the middle and older population, and good control of hypertension may reduce the LBP burden. Special attention is needed to prevent LBP in women. Further study with a larger sample size addressing these neglected issues may have more clarifications to decrease the burden of LBP.” (page 13, lines 443–449)
14.	I suggest providing the STROBE checklist for cross-sectional studies.	We had included the STROBE checklist during our initial submission in RMD Open (20 October 2021) which was then passed on to BMJ Global Health (3 November 2021) and later on to BMJ Open (10 November 2021). However, we do not see the document now in the current list of submitted files in BMJ Open. Since the manuscript is revised, we submit the revised STROBE checklist as a separate file named: “STROBE_cklst_LBP_bmjopen-2021-059192_29Jun2022.docx”

VERSION 2 – REVIEW

REVIEWER	Shabbir Sany Faridpur Medical College, Community Medicine
REVIEW RETURNED	17-Jul-2022
GENERAL COMMENTS	Thank you for giving me the opportunity to review the revised version of this interesting manuscript. The authors have addressed all the issues that were raised earlier. In this present form, the manuscript can be accepted for publication.
REVIEWER	Aleksander Galaś Jagiellonian University, Department of Epidemiology and Preventive Medicine
REVIEW RETURNED	25-Jul-2022

GENERAL COMMENTS	Dear Editor, Dear Authors, The manuscript has been nicely improved after revision. I have to highlight, however, an important issue, which was not corrected. This refers to "risk factor" concept. The risk factor in epidemiology and medicine implies the factor which precedes the outcome. Although there are some discussion about calling a factor as a risk factor based on a cross-sectional investigation, this strategy is not appropriate in general (see book Modern Epidemiology by KJ Rothman, S Greenland, or others). The fact that some other cross-sectional based publications used this term does not justifies using it by other Authors. Referring to some details ... in the submitted manuscript the Authors identified hypertension but not obesity as a risk factor. The first one is more discussible, the second clearly better evident and understandable (referring to the available research results and to known pathologic mechanisms). Therefore, I still argue for using "factors associated with LBP" instead of "risk factors of LBP" in the submitted manuscript. Hopefully, this will be taken into consideration, AG
--

VERSION 2 – AUTHOR RESPONSE

Reviewer 3: Dr. Aleksander Gałaś		Author response
a.	The manuscript has been nicely improved after revision.	Thank you for your appreciation.
b.	I have to highlight, however, an important issue, which was not corrected. This refers to "risk factor" concept. The risk factor in epidemiology and medicine implies the factor which precedes the outcome. Although there are some discussion about calling a factor as a risk factor based on a cross-sectional investigation, this strategy is not appropriate in general (see book Modern Epidemiology by KJ Rothman, S Greenland, or others). The fact that some other cross-sectional based publications used this term does not justifies using it by other Authors. Referring to some details ... in the submitted manuscript the Authors identified hypertension but not obesity as a risk factor. The first one is more discussible, the second clearly better evident and understandable (referring to the available research results and to known pathologic mechanisms). Therefore, I still argue for using "factors associated with LBP" instead of "risk factors of LBP" in the submitted manuscript. Hopefully, this will be taken into consideration.	Although there are views to the contrary, we agree with you to change the term from "risk factors" to "factors". We have revised the text throughout the manuscript. Page 2, line 56; Page 3, lines 80, 84 and 88; Page 5, line 135; Page 9, line 289; Page 10, line 317; Page 13, lines 397 and 398; and Page 19, line 453;